# Topology aware multitask cascaded U-Net for cerebrovascular segmentation

**Pierre Rougé** [1,2] *, **Nicolas Passat** [1], **Odyssée Merveille** [1]

**1** CReSTIC EA 3804, Université de Reims Champagne Ardenne, Reims, France, **2** CNRS, Inserm, CREATIS UMR 5220, U1294 INSA-Lyon, Université Claude Bernard Lyon 1, UJM-Saint Etienne, Univ Lyon, Lyon, France

* pierre.rouge@creatis.insa-lyon.fr

**Data Availability Statement:** Among the two dataset used in this study, Bullitt is publicly available, including both annotations and TOF-MRA images (available at: https://public.kitware.com/Wiki/TubeTK/Data). For IXI, only the TOF-MRA

## Abstract

Cerebrovascular segmentation is a crucial preliminary task for many computer-aided diagnosis tools dealing with cerebrovascular pathologies. Over the last years, deep learning based methods have been widely applied to this task. However, classic deep learning approaches struggle to capture the complex geometry and specific topology of cerebrovascular networks, which is of the utmost importance in many applications. To overcome these limitations, the clDice loss, a topological loss that focuses on the vessel centerlines, has been recently proposed. This loss requires computing the skeletons of both the manual annotation and the predicted segmentation in a differentiable way. Currently, differentiable skeletonization algorithms are either inaccurate or computationally demanding. In this article, it is proposed that a U-Net be used to compute the vascular skeleton directly from the segmentation and the magnetic resonance angiography image. This method is naturally differentiable and provides a good trade-off between accuracy and computation time. The resulting cascaded multitask U-Net is trained with the clDice loss to embed topological constraints during the segmentation. In addition to this topological guidance, this cascaded U-Net also benefits from the inductive bias generated by the skeletonization during the multitask training. This model is able to predict the cerebrovascular segmentation with a more accurate topology than current state-of-the-art methods and with a low training time. This method is evaluated on two publicly available time-of-flight magnetic resonance angiography (TOF-MRA) images datasets, also the codes of the proposed method and the reimplementation of state-of-the-art methods are made available at: https://github.com/PierreRouge/Cascaded-U-Net-for-vessel-segmentation.

## 1 Introduction

Vascular diseases encompass various alterations in blood vessels (e.g. stenosis, aneurysm, thrombosis, or embolism) with significant consequences such as stroke, a leading cause of death and disability. The accurate vessel segmentation from angiographic images, actively investigated over the last 30 years [1, 2], is a crucial step for the diagnosis and treatment of vascular diseases. Over the last decade, deep learning has allowed significant progress in medical

images are publicly available (https://brain-development.org/ixi-dataset/), and we used annotations provided by a third party which can shared the annotations on demand (contact: zuluaga@eurecom.fr).

**Funding:** This work was supported by the French Agence Nationale de la Recherche (Grant ANR-20-CE45-0011). The funders had no role in study design, data collection and analysis, decision to publish, or preparation of the manuscript.

**Competing interests:** The authors have no relevant financial or non-financial interests to disclose.

imaging and especially in segmentation. State-of-the-art approaches predominantly rely on architectures resembling U-Net [3–5] or transformer-based models [6–8]. Despite these efforts, automatic segmentation of vascular networks remains a challenging issue, especially due to the complex topological and geometrical properties of vessels, and their sparseness in the images. By contrast to many anatomical structures, vessels do not constitute a compact volume at a specific position and scale. They are organized as a multiscale network (from large vessels to thin ones close to / beyond the resolution of the acquisition) in the whole image. This represents a challenge for deep learning methods, especially when a topologically correct result is required for subsequent tasks (e.g. blood flow modeling [9]).

To overcome this challenge, Shit et al. [10] recently proposed a novel metric specifically designed to evaluate the quality of tubular structure segmentation. This metric, named clDice (for "centerline Dice"), mainly relies on the skeleton of the tubular structures instead of their whole volume, therefore focusing on topological information. To use this new metric as a loss function, it is necessary to compute the skeleton of the predicted segmentation in a differentiable manner. Therefore, the authors proposed a differentiable soft-skeleton algorithm. However, the resulting skeletons do not preserve the topology of the structures of interest. In a subsequent work, Menten et al. [11] proposed two new differentiable skeletonization algorithms to overcome this limitation. They showed that using the clDice loss provides better and more connected segmentation of 2D tubular structures, for instance on retinal images and on 3D tubular structures of the Vessap dataset [12] (a dataset of mice brain vascular networks acquired at a very high resolution and with a research protocol). However, such approach has not yet been tested on Magnetic Resonance Angiography (MRA) or X-ray Computed Tomography Angiography (CTA) datasets acquired in clinical conditions (images with more noise, artifacts and with a lower resolution).

In this article, a cascaded network with a U-Net backbone [3] is proposed, which first computes the segmentation and then uses this segmentation and the initial image to perform the skeletonization task. In this architecture, the skeletonization task directly benefits from the segmentation, and may also incorporate information from the initial image to produce a better skeleton. Finally, the skeleton output, obtained in a differentiable way, is used to compute the clDice loss and supervise the whole network, resulting in a more topologically correct segmentation.

The method is evaluated against four standard U-Net models, trained with either the Dice or clDice losses using the skeletonization methods introduced in [10, 11]. Additionally, two other methods aimed at preserving topology in vascular segmentation [13, 14], which have not yet been tested for cerebrovascular segmentation, are included in this evaluation, along with two well-established architectures for vascular segmentation [15, 16]. This study demonstrated that the proposed method provides segmentations with a more accurate topology while having a lower training time.

The main contributions of this article are the following:

- The performance of the clDice loss and state-of-the-art methods for 3D cerebrovascular segmentation are evaluated;

- The code for all compared methods is provided within a unified PyTorch framework, designed to be easily used and extended by the community for benchmarking 3D brain vascular segmentation: https://github.com/PierreRouge/Cascaded-U-Net-for-vessel-segmentation;

- An efficient way of performing the skeletonization operation to compute the clDice is proposed;

- A cascaded multitask U-Net architecture is proposed, which segments brain vascular networks with a more accurate topology and with a lower training time.

The remainder of the article is organized as follows. In Section 2, the state-of-the-art of cerebrovascular segmentation is discussed. In Section 3, the methodological contribution is described. In Section 4, experiments wich compare the method with state-of-the-art ones are presented. In Section 5, these methods and results are discussed. Section 6 concludes this article with perspective works.

## 2 Related works

In this section, recent deep learning methods for vascular segmentation are discussed. The vesselness filters techniques are not covered (see e.g. [17] for a recent survey), and the focus is mainly—though not exclusively—on methods applied to cerebrovascular networks.

### 2.1 Deep learning for vascular segmentation

Deep learning has seen extensive applications in cerebrovascular segmentation, as evidenced by the work of Chen et al. [18]. Pioneering efforts in the field include those by Livne et al. [19] and Sanches et al. [20] who leveraged architectures based on both 2D and 3D U-Net. Notably, Sanches et al. augmented the conventional U-Net architecture with inception modules to enhance the network representational capacity.

Subsequent efforts were directed toward designing networks specifically tailored for curvilinear structure segmentation. For instance, Mou et al. [15] introduced a U-Net-like convolutional network equipped with two attention modules to capture both spatial and channel relationships. Additionally, they employed $1 \times 3$ and $3 \times 1$ kernels to better capture boundary features in various spatial directions, demonstrating its efficacy across multiple imaging modalities and datasets.

Similarly, Ni et al. [21] proposed incorporating channel attention during the aggregation of low and high-level features in the decoder phase. They also integrated an Atrous Spatial Pyramid Pooling (ASPP) module into the bottleneck to augment the receptive field of their architecture, yielding promising results on a private CTA dataset.

In parallel, Tetteh et al. [16] devised an architecture aimed at simultaneous vessel segmentation, centerline prediction, and bifurcation detection in angiographic images. Their architecture, a fully convolutional network (FCN) with four convolutional layers and a final classification layer, stands out for its use of a 3D convolution with a cross-hair filter, enabling to capture 3D information without excessive computational overhead. Notably, the absence of pooling operations in this FCN architecture helps to preserve small vessel structures in the feature maps, thereby enhancing segmentation performance.

In a similar vein, Guo et al. [22] mitigated computational complexity while retaining 3D spatial information by combining three U-Net networks trained on 2D slices in orthogonal directions, resulting in a 2.5D U-Net.

Furthermore, Xia et al. [23] introduced a method emphasizing the significance of edge voxels. Their approach incorporates a reverse edge attention module to refine features in skip connections by accentuating edge information in the feature maps. Additionally, they propose a novel loss term to impose stricter constraints on prediction boundaries.

Recently, Valderrama et al. [24] proposed to incorporate the skull-stripping step directly in a multitask architecture. In addition, they used free adversarial training (gradient based perturbation on the input data) to compensate the lack of annotated data. They demonstrate that their model achieve competitive results on two cerebrovascular segmentation datasets.

Also, Dang et al. [25] addressed the challenge of data annotation by proposing a weakly-supervised deep learning framework. Annotated patches were generated using a classifier to distinguish vessel from non-vessel patches and the K-means algorithm.

While these methods excel in voxel-wise segmentation, they may fall short in accurately capturing the complex geometry and topology of cerebrovascular structures.

## 2.2 Topology aware segmentation

In recent developments, several methods have emerged to address the challenge of incorporating topological priors or constraints during training. One notable example is the work by La Barbera et al. [13], which introduces a loss function based on vesselness. Specifically, this loss function comprises a term comparing the eigenvalues of the Hessian matrix and another term enforcing a high Frangi vesselness value for voxels within the manual annotation. The authors demonstrated the effectiveness of this approach in improving the segmentation of arteries, veins and ureters in contrast-enhanced Computed Tomography (ceCT) scanners.

Keshwani et al. [26] proposed a multitask architecture featuring a shared encoder and three distinct decoders. These decoders include a standard segmentation decoder, a decoder outputting a distance map to identify vessel centers, and a decoder outputting a vector for each voxel. Following this, a calculation of $L_2$ distance is performed between the vectors produced by the last decoder, considering two central voxels. Then the network is trained to output vectors with a $L_2$ distance proportional to the topological distance if the voxels belong to the same vascular tree and with a high $L_2$ distance otherwise. After training, this learned distance is used to construct the vascular tree starting from vessel sources using Dijkstra multi-source shortest path tree algorithm. This method has the advantage of naturally building a fully connected vascular network, however the latter is not a segmentation but a skeleton. Furthermore, while primarily relevant for multitask segmentation tasks such as portal/hepatic vein or artery/vein segmentation, this architecture presents a novel approach to incorporate topological information.

Similarly, Wang et al. [14] introduced a multitask architecture outputting both segmentation and distance maps. First, a thinning is applied to the probability map to obtain a binary skeleton and then the real radius is obtained by fitting Gaussian kernels to each voxel with the standard deviation based on the value of the voxel in the predicted distance map. This process results in smoother and more accurate edges compared to voxel-wise segmentation methods.

A significant progress comes from Shit et al. [10], who introduced the centerline Dice (clDice) loss function based on the segmentation skeleton, thereby avoiding bias towards large vessels. This method involves computing the skeleton of the predicted segmentation in a differentiable manner using a soft-skeleton algorithm proposed by the authors. Additionally, Menten et al. [11] proposed two new differentiable skeletonization methods.

Stucki et al. [27] proposed a topological loss function based on persistent homology, tested on various 2D datasets with different topological characteristics. While some datasets feature curvilinear, non-vessel, structures, this approach represents a novel exploration of topological considerations.

Furthermore, some works leverage discrete Morse theory (DMT) to identify topologically significant structures. For instance, Hu et al. [28] employ a loss function focusing on detected Morse structures to ensure the correct segmentation of these critical structures. Additionally, Gupta et al. [29] utilize DMT to compute a meaningful uncertainty map, that can be used to improve the segmentation topology through a semi-automatic post-processing workflow.

The recent TopCoW challenge [30] provides a comprehensive overview of the current state-of-the-art in vessel segmentation. Participants predominantly employed U-Net

architectures (mostly nnU-Net) with 3D patches, and the use of data augmentation and ensembling significantly increased performance. However, only a few teams utilized loss functions dedicated to vascular segmentation, but those that did achieved better results, particularly in terms of segmentation topological correctness. This highlights the effectiveness and importance of approaches using topology aware loss functions for cerebrovascular segmentation.

## 3 Method

In this section, the clDice and the different skeletonization algorithms are presented (Section 3.2). Then, the proposed cascaded multitask U-Net architecture is described (Section 3.3). Finally, the details of the training configuration of the architecture are provided (Section 3.4).

### 3.1 Compliance with ethical standards

This research study was conducted retrospectively using human subject data made available in open access by Kitware at the following link: https://public.kitware.com/Wiki/TubeTK/Data and the Imperial College London at https://brain-development.org/ixi-dataset/. Ethical approval was not required, as confirmed by the license attached with the open access data.

### 3.2 clDice loss and differentiable skeletonization

The clDice [10] derives from two metrics called *topology precision* ($T_{prec}$) and *topology sensitivity* ($T_{sens}$) in reference to the usual precision and sensitivity metrics. These metrics are defined as follows:

$$T_{prec}(C_P, S_G) = \frac{|C_P \cap S_G|}{|C_P|}, \tag{1}$$

$$T_{sens}(C_G, S_P) = \frac{|C_G \cap S_P|}{|C_G|}, \tag{2}$$

where $C_P$, $C_G$ and $S_P$, $S_G$ are the predicted and manually annotated centerlines and segmentations, respectively. The clDice is defined as the harmonic mean of $T_{prec}$ and $T_{sens}$:

$$clDice(S_P, S_G, C_P, C_G) = 2 \cdot \frac{T_{prec}(C_P, S_G) \cdot T_{sens}(C_G, S_P)}{T_{prec}(C_P, S_G) + T_{sens}(C_G, S_P)}. \tag{3}$$

By leveraging the skeleton representation, the clDice avoids being biased by large vessels and thus better focuses on topological information. However, most of the methods designed to extract a skeleton are not differentiable. Therefore, Shit et al. proposed a differentiable soft-skeleton algorithm to use the clDice for training a neural network. This algorithm uses min and max filters to perform dilation and erosion on the predicted segmentation. Preliminary experiments (see Section 4.3) showed that the results from this soft-skeletonization are not sufficiently accurate for 3D vascular segmentation, in particular regarding topology.

In a subsequent work, Menten et al. [11] proposed two new differentiable skeletonization algorithms. These algorithms remove simple points [31] in the image, ensuring that the topology is not affected by the skeletonization. The identification of simple points is done either using the Euler characteristics or through a Boolean characterization. In the following, these methods will be referred to as Euler and Boolean methods, respectively. These methods have the advantage of generating a nearly topologically correct skeleton, but at the cost of a high computation time (see Section 4.3).

A standard U-Net was chosen to perform the skeletonization in this work. By nature, this method is differentiable and provides a good trade-off between accuracy and computation time.

### 3.3 Model architecture

The backbone model used in the proposed method (prop. meth.) is a standard U-Net [3] with a depth of 4, using 2-stride convolution for down-sampling, instance normalization and leakyReLU activation function (see Fig 1).

Our cascaded U-Net architecture is presented in Fig 2. It is composed of a first U-Net taking as input an MRA image and performing the segmentation. This task is supervised by a Dice loss and will be referred to as the *segmentation network*. The output of this network is

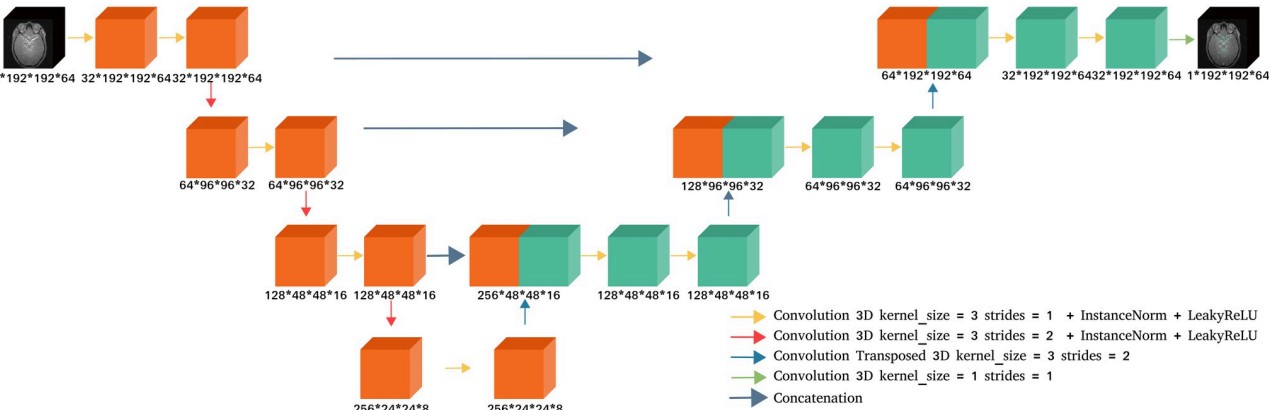

**Fig 1. The baseline U-Net architecture used in the proposed approach (see Section 3.3).** The output of this network is either the vascular segmentation (as shown here) or the vascular skeleton, depending on the chosen task.

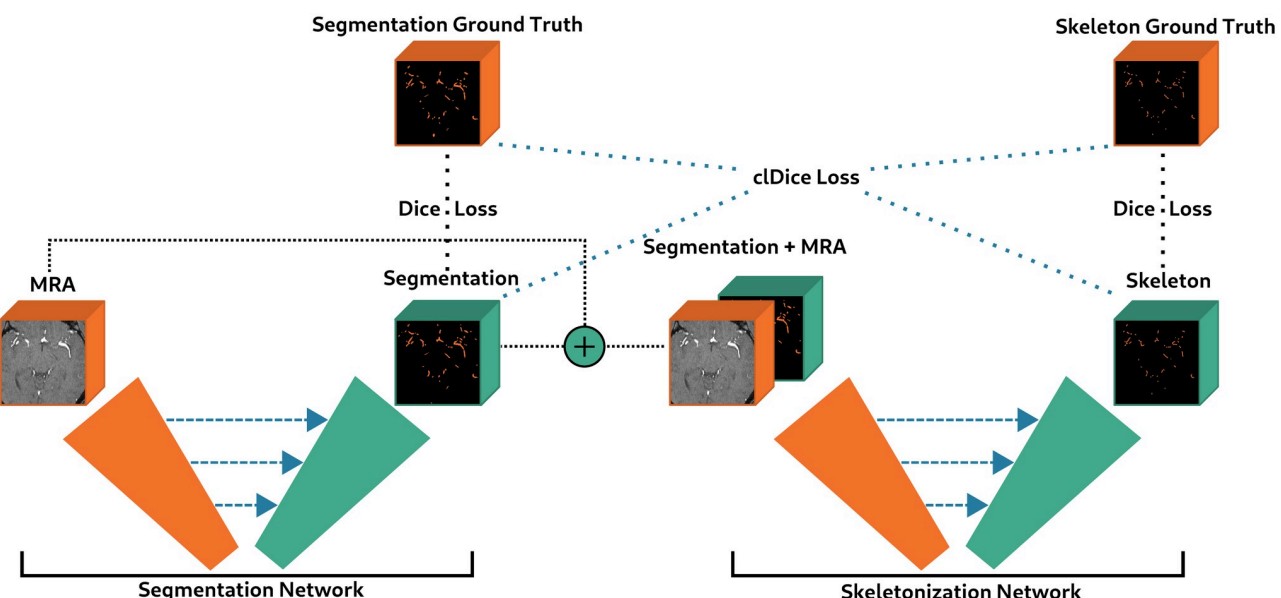

**Fig 2. Architecture of the proposed cascaded U-Net (see Section 3.3).**

concatenated with the MRA image and fed to a second U-Net performing the skeletonization task, also supervised by a Dice loss. This part of the architecture will be referred to as the *skeletonization network* (skel. network). The training of these two networks is also supervised by the clDice loss, which takes as input the predicted segmentation and the predicted skeleton. The cascaded U-Net final loss is then defined by:

$$Loss(S_P, S_G, C_P, C_G) = \begin{cases} Dice(S_P, S_G) \ + \\ \lambda_1 \cdot Dice(C_P, C_G) \ + \\ \lambda_2 \cdot clDice(S_P, S_G, C_P, C_G), \end{cases} \tag{4}$$

where $\lambda_1, \lambda_2 \in \mathbb{R}$ are two weight parameters.

This architecture presents several advantages. First, the skeletonization network performs the skeletonization in a differentiable manner, which allows using the clDice loss and enforcing topological constraint on the segmentation task. Second, in contrast to other skeletonization methods, the proposed skeletonization network takes as input the original MRA image, enabling the correction of small segmentation errors and thus limiting error propagation. Finally, by jointly learning segmentation and skeletonization, the segmentation network benefits of an inductive bias that encourages learning topologically correct segmentations. Indeed, the skeletonization task gives similar importance to all vessels independently of their thickness; so the multitask learning can help to enforce the influence of small vessels in the segmentation task.

### 3.4 Training configuration

All the MRA volumes were first normalized by Z-score. During training, one batch is composed of 2 patches of size $192 \times 192 \times 64$, each randomly located in an MRA volume. One epoch consists of 100 batches.

To ensure a fair comparison across the experiments, the same data augmentation strategy (inspired by nnU-Net [4]) was used for all trained networks using the Python package `batchgenerators` [32]:

- **Rotation**—Applied around each axis ($x$, $y$, $z$) with a probability of 0.2. The angles of rotation are drawn from a uniform distribution $U(-30, 30)$.

- **Scaling**—Applied with a probability of 0.2. Factor drawn from $U(0.7, 1.4)$.

- **Gaussian noise**—Applied with a probability of 0.1. Variance is drawn from $U(0.0, 0.1)$.

- **Gaussian blur**—Applied with a probability of 0.1. The width of the Gaussian kernel is drawn from $U(0.5, 1.0)$.

- **Brightness**—Modify the voxel intensities by a multiplicative factor with a probability of 0.15. The multiplicative factor is drawn from $U(0.75, 1.25)$.

- **Contrast**—Modify the voxel intensities by a multiplicative factor and clip them to the original range value, with a probability of 0.15. The multiplicative factor is drawn from $U(0.75, 1.25)$.

- **Simulation of low resolution**—Downsample the image with nearest neighbour interpolation, then upsample it to its original size with cubic interpolation, with a probability of 0.125. Downsampling factor is drawn from $U(0.5, 1.0)$.

- **Gamma transform**—The input is normalized in $[0, 1]$; then the voxel intensity $i$ is transformed, with probability of 0.1, as follows: $i_{new} = i_{old}^{\gamma}$, with $\gamma \sim U(0.7, 1.5)$.

- **Mirroring**—Patches are mirrored along each axis.

Stochastic gradient descent with Nesterov momentum was used, with an initial learning rate set to 0.01. A linear learning rate decay was applied, resulting in the learning rate being equal to 0 at the last epoch.

During inference, sliding windows with a 25% overlap were employed to reconstruct the full volume. To mitigate artifacts at overlapping regions, a Gaussian kernel was applied to reduce the weight of voxels further from the center of the patches. Test time augmentation was implemented by applying flips with respect to the three axes to the input volume, and all resulting probability maps were averaged before thresholding. Subsequently, a post-processing step was applied to remove all connected components smaller than 100 voxels from the segmentation outputs of all methods.

For the cascaded multitask U-Net, the two U-Nets were first pretrained separately before fine-tuning the network as described in Section 3.3. The weights of the segmentation network were initialized with the weights of the U-Net trained with the Dice loss, and the skeletonization network was pretrained using the MRA image and the segmentation annotation as input. All models were trained for 500 epochs; except for the cascaded multitask U-Net, which was first pretrained during 500 epochs and then fine-tuned for 250 epochs.

## 4 Evaluation

The setup and results of the experiments conducted to evaluate the skeletonization network and the cascaded multitask U-Net are presented in this section. All results for deep learning models were obtained through a 5-fold cross-validation.

### 4.1 Datasets

For this study, the publicly available Bullitt dataset [33], which contains 34 time-of-flight magnetic resonance angiography (TOF-MRA) volumes of the brain, was used. All the volumes present a voxel resolution of $0.513 \times 0.513 \times 0.800$ mm$^3$ and a size of $448 \times 448 \times 128$. Each volume was annotated by one expert. To produce the skeleton annotations, the `skeletonize` function from the Python package `scikit-image`, which implements the gold-standard Lee algorithm [34], was used.

The publicly available IXI dataset, which initially comprises approximately 600 TOF-MRA images acquired from three distinct centers, was also used. The focus was specifically on the subset from Guy's Hospital, London, UK, which consists of 316 volumes and 15 annotations. However, only the 15 annotated volumes were used in this study, as all methods are fully supervised. All volumes maintain uniform resolution, with voxels measuring $0.47 \times 0.47 \times 0.80$ mm$^3$ and dimensions of $512 \times 512 \times 100$.

An illustration of these two datasets is presented in Fig 3.

### 4.2 Metrics

In the evaluation, the clDice (see Section 3.2) and the Dice similarity coefficient (*DSC*) [35] defined below, were used.

$$DSC = \frac{2 \cdot tp}{2 \cdot tp + fp + fn},$$ (5)

where *tp*, *fp*, *fn* are the true positives, false positives and false negatives, respectively. Both *clDice* and *DSC* take values in the range [0, 1] and are without units.

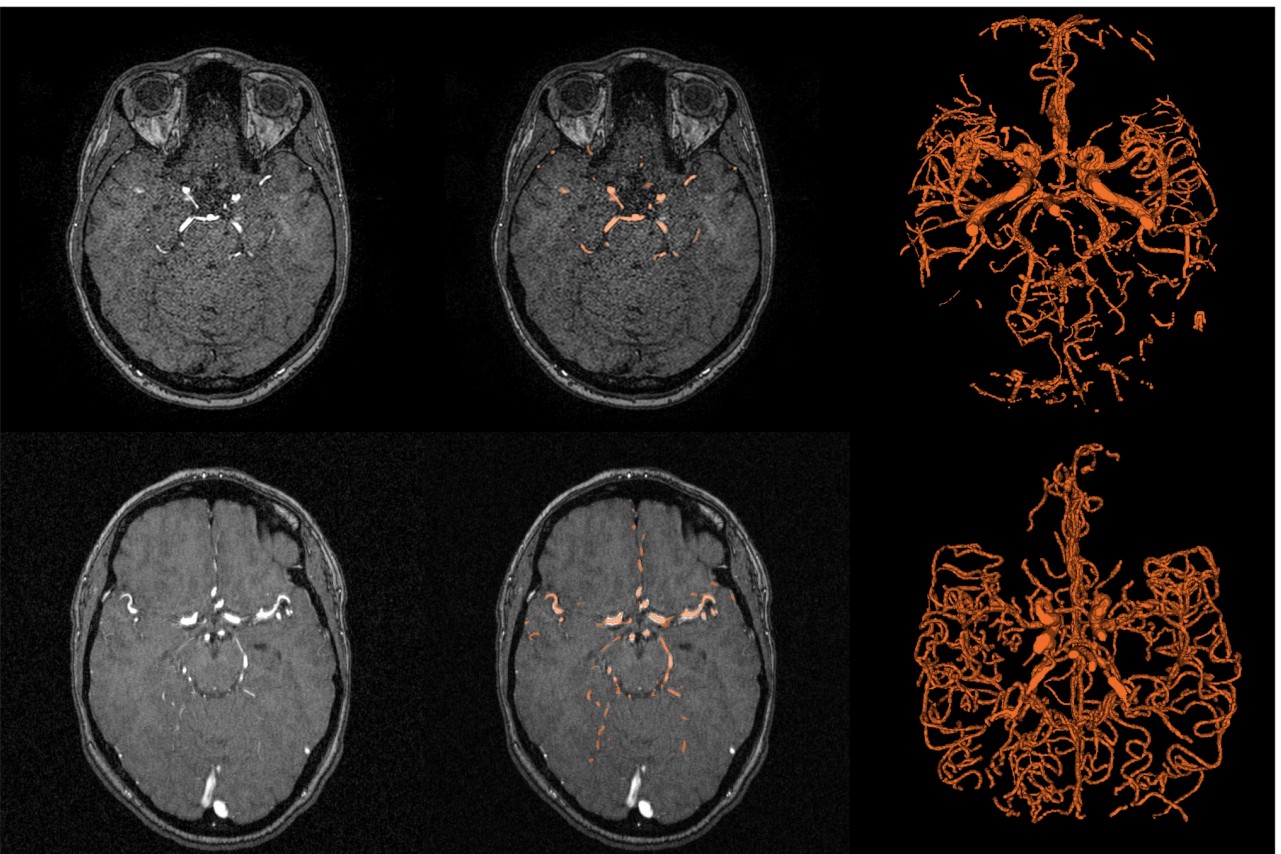

**Fig 3.** From left to right: a slice of the TOF-MRA volume, the same slice superimposed with its manual annotation (in transparent orange) and the 3D volume of the manual annotation. First row: taken from patient IXI017 of the IXI dataset. Second row: taken from Patient 8 of the Bullitt dataset.

Two distance metrics were also computed: the average symmetric surface distance metric (*ASSD*) and the Hausdorff Distance 95% percentile (*HD*95), defined as:

$$ASSD(P, R) = \frac{\sum_{p \in P} d(p, R) + \sum_{r \in R} d(r, P)}{|P| + |R|}, \tag{6}$$

$$HD95(P, R) = \max\{d_{95}(P, R), d_{95}(R, P)\}, \tag{7}$$

with:

$$d_{95}(A, B) = x_{95}\{\min_{b \in B} d(a, b)\}, \tag{8}$$

$$d(a, B) = \min_{b \in B} d(a, b), \tag{9}$$

where $P$ is the predicted segmentation and $R$ the reference, $d(a, b)$ is the Euclidean distance between voxels $a$ and $b$, and $x_{95}$ denotes the 95% percentile. Both $HD95$ and $ASSD$ are expressed in millimeters (mm).

Beyond these quantitative metrics, the topological quality of the segmentation was also evaluated by computing topological descriptors: the first Betti number $\beta_0$ (i.e. the number of

**Table 1. Results of the proposed skeletonization network vs. other skeletonization methods.**

| Model | Runtime (ms) ↓ | $\beta_0$ ↓ | | $\beta_1$ ↓ | |
|---|---|---|---|---|---|
| | | **Bullitt** | **IXI** | **Bullitt** | **IXI** |
| Ground-truth | | 29 ± 12 | 100 ± 50 | 150 ± 30 | 99 ± 34 |
| Soft-skeleton algorithm | 5 ± 1 | 1197 ± 245 | 1590 ± 276 | 6 ± 3 | 17 ± 7 |
| Euler | 558 ± 13 | 30 ± 13 | 121 ± 54 | 151 ± 30 | 108 ± 34 |
| Boolean | 1022 ± 34 | 29 ± 13 | 113 ± 55 | 151 ± 30 | 99 ± 34 |
| Skeletonization network | 9 ± 2 | 294 ± 48 | 610 ± 206 | 118 ± 26 | 17 ± 7 |

connected components), the second Betti number $\beta_1$ (i.e. the number of tunnels/cycles), and the third Betti number $\beta_2$ (i.e. the number of cavities).

Unlike other metrics, Betti numbers characterize the topology of a structure independently of its proposed annotation. Analyzing the Betti numbers of a segmentation results then requires comparing them with the true (ground truth) Betti numbers.

From an anatomical point of view, the topology of the brain arterial network is well established. All arteries are connected, thus $\beta_0$ is equal to 1; there is one tunnel (the circle of Willis), thus $\beta_1$ is equal to 1; and no cavities are present, thus $\beta_2$ is equal to 0. However, the Betti numbers of the Bullitt and IXI annotations were computed, and much greater values for $\beta_0$ and $\beta_1$ were observed (see Table 1). This discrepancy is a typical issue with segmentation annotations of geometrically complex 3D structures. Manual annotation is a labor-intensive task, usually performed on 2D slices, making it difficult to accurately capture the 3D geometry and topology of the annotated structure.

Moreover, the annotation guidelines rarely focus on topology, as voxel-wise metrics are the gold standard for segmentation evaluation. This leads to annotations that are suitable for voxel-wise analysis but often inadequate for topological analysis. Our goal is to obtain vascular segmentation that reflects the true topology of the brain arterial network. In the following, the ground-truth Betti numbers of the brain artery network ($\beta_0 = 1$, $\beta_1 = 1$, and $\beta_2 = 0$) are used as the reference for the segmentation Betti numbers.

Additionally, the value of $\beta_2$ was computed for both Bullitt and IXI datasets, it was observed that the values were null for both manuals annotations and predicted segmentations (corresponding to the theoretical value). Therefore, the $\beta_2$ values are not presented or discussed in the following sections.

The statistical significance between every method and for every metric was also evaluated. A t-Test was conducted if a normal distribution was followed by the data, and a Wilcoxon test was used otherwise. The normality of the metrics was tested using a Shapiro test.

## 4.3 Skeletonization using U-Net

The skeletonization network was compared to (1) the soft-skeleton algorithm introduced in [10], (2) the skeletonization based on the Boolean characterization of simple points and (3) the skeletonization based on the Euler characteristic (i.e. the alternating sum of the Betti numbers, see e.g. [11]). For all methods, the mean time required to perform the skeletonization per patch and the topological metrics introduced in Section 4.1 were computed.

The results, presented in Table 1, demonstrate that the proposed method yields skeletons with a more accurate topology than the soft-skeleton algorithm in a comparable computation time. Euler and Boolean methods produce nearly perfect skeletons when compared to the skeletons ground truths but at the cost of a more important runtime. Our skeletonization network

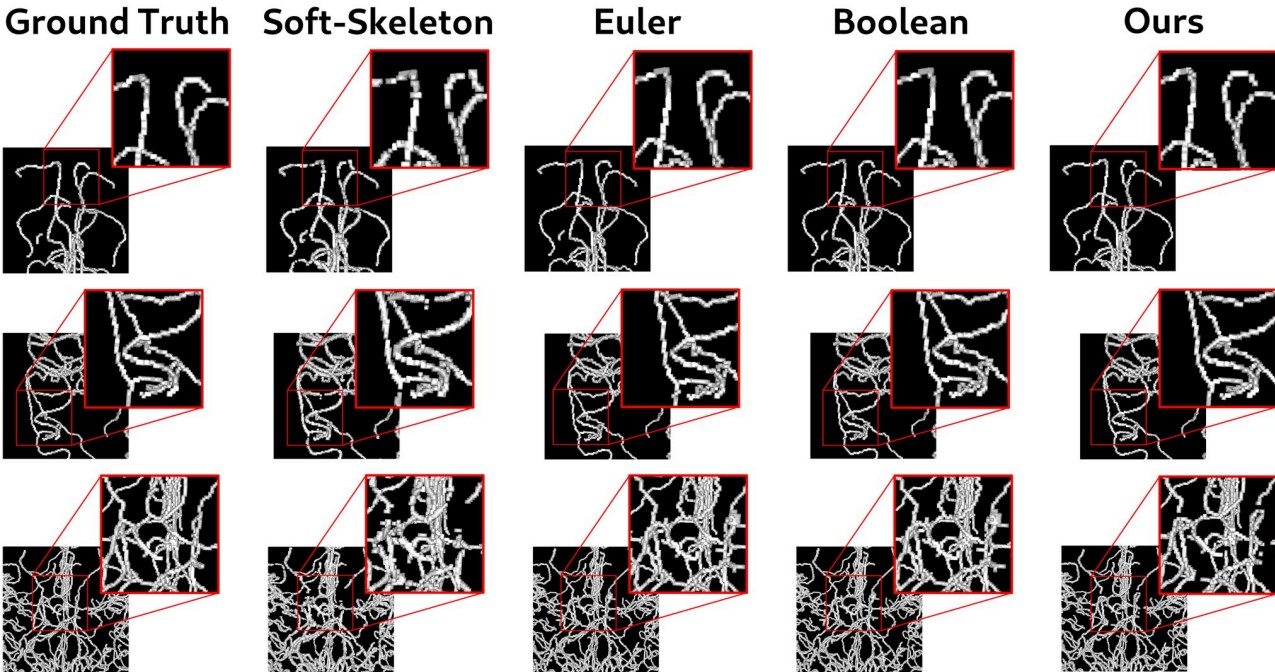

**Fig 4. Comparison of the skeletonization methods.** The skeletons generated by the soft-skeleton algorithm exhibit major disconnections and a thickness spanning several voxels. Conversely, those produced by the proposed method (last column) display slight disconnections but maintain a correct thickness of one voxel.

then provides a good trade-off between accuracy and computation time. We draw the reader's attention to the values of $\beta_1$. It appears that the soft-skeleton algorithms and the proposed method reduce the number of tunnels. However, the actual meaning of these tunnels remains questionable regarding their anatomical / noisy nature. Consequently, the above discussion is mostly related to the value of $\beta_0$, evaluating the connectivity of the skeletons. The $\beta_1$ values are presented for the sake of completeness.

Qualitatively, the skeletons produced by the soft skeleton algorithm present many disconnections and a thickness of several voxels compared to other skeletonization methods. By contrast, the proposed method provides more accurate skeletons (see Fig 4). Visually, it appears that the gap in topological metrics between the proposed method and Euler and Boolean methods is primarily due to minor disconnections. Overall, the skeletons appear quite similar.

## 4.4 Cascaded multitask U-Net

**4.4.1 Hyperparameters optimization.** The goal of the cascaded multitask U-Net is to improve the results of the segmentation network thanks to the clDice loss. As stated in Section 3.3, the hyperparameters $\lambda_1$ and $\lambda_2$ have to be set in order to handle the trade-off between the skeletonization loss and the clDice loss. Two training configurations were also tested: one in which the skeletonization network weights are frozen, and another in which they are updated during the cascaded U-Net training. In both configurations, the skeletonization network was first pre-trained. Therefore, a grid search was performed to select these parameters. Based on these experiments, it was found that the best cascaded multitask U-Net training policy consists of freezing the weights of the skeletonization network and setting the loss weights to $\lambda_1 = \lambda_2 = 0.5$.

**4.4.2 Baseline methods for comparison.** To assess the performance of the cascaded U-Net, comparisons were made with several state-of-the-art methods, categorized into three groups. First, methods using the clDice metric but employing different skeletonization algorithms, as presented in Section 3.2, were considered. Second, methods with topological priors, including the Tubular Structures Loss Function (*TsLoss*) from La Barbera et al. [13] and the DeepDistanceTransform method from Wang et al. [14], were examined. Third, two well-established vascular segmentation architectures, namely CS2-Net [15] and DeepVesselNet [16], were considered.

All these methods were reimplemented with the help of the respective git repositories in a common Pytorch framwork. (We thank the authors for having made these codes available).

For all methods, the same U-Net backbone model, as described in Section 3.3, was used, except for CS2-Net [15] and DeepVesselNet [16], where the proposed architectures were employed. For the DeepDistanceTransform model [14], a second decoder branch was introduced into the baseline U-Net solely for distance map prediction.

The initial learning rate was decreased to 0.001 for CS2-Net [15] due to instability in the training caused by vanishing gradients.

For the Deep Distance Transform method [14], the geometry-aware refinement (GAR) introduced in the same work was used to produce the final segmentation. Additionally, the cross-entropy loss used for the distance map prediction was weighted with the respective proportion of each class; otherwise, the background class was too preponderant.

Considering hyperparameter optimization, a grid search was performed to select the optimal value of the clDice weight for each method using clDice loss. The parameters of the reparametrization trick, which enable the differentiable binarization required for the Euler and Boolean methods, were set to $\beta = 0.33$ and $\tau = 1.0$.

Regarding the method of La Barbera et al. [13], the following values indicated in the original paper were used: $w_{ms} = 0.05$ for the *MsLoss* weight, and $\alpha = 0.1$, $\beta = 0.1$, $\gamma = 2$ for the Frangi parameters. The maximal size of Gaussian kernels applied to the predicted segmentation and the manual annotation before computing the Hessian matrix was set to $\sigma_{max} = 15$.

**4.4.3 Results.** The experiments conducted on the Bullitt dataset are summarized in Table 2. Boxplots indicating statistics and statistical significance are presented in Figs 5 and 6. Firstly, the proposed method outperforms CS2-Net [15], DeepVesselNet [16] and DeepDistanceTransform [14] in terms of overlap-based metrics, $\beta_0$, and distance-based metrics (statistically significant). This demonstrates its competitiveness compared to state-of-the-art methods. The proposed approach yields a relatively higher $\beta_1$, comparable to the topological Boolean and Euler approaches.

**Table 2. Evaluation of the presented methods on Bullitt dataset: Mean ± standard deviation values.**

| Model | DSC ↑ | clDice ↑ | ASSD ↓ | HD95 ↓ | $\beta_0$ ↓ | $\beta_1$ ↓ | Training time (h) ↓ |
|---|---|---|---|---|---|---|---|
| U-Net (Dice) | **0.76** ± 0.02 | **0.85** ± 0.02 | 0.94 ± 0.13 | 7.20 ± 1.05 | 26.4 ± 4.6 | 116.0 ± 27.8 | 17 h |
| U-Net (Dice + clDice Soft) [10] | 0.75 ± 0.02 | **0.85** ± 0.02 | **0.92** ± 0.13 | **6.84** ± 1.11 | 27.7 ± 5.6 | 122.7 ± 28.3 | 18 h |
| U-Net (Dice + clDice Euler) [11] | 0.75 ± 0.02 | **0.85** ± 0.02 | **0.92** ± 0.14 | 7.12 ± 0.97 | 26.9 ± 5.5 | 132.0 ± 27.7 | 28 h |
| U-Net (Dice + clDice Boolean) [11] | **0.76** ± 0.01 | **0.85** ± 0.02 | **0.92** ± 0.12 | 7.07 ± 1.15 | 23.4 ± 5.4 | 131.4 ± 30.1 | 42 h |
| DeepVesselNet [16] | 0.71 ± 0.02 | 0.81 ± 0.02 | 1.17 ± 0.15 | 9.27 ± 1.81 | 55.9 ± 7.8 | 112.1 ± 27.1 | 10 h |
| CS2-Net [15] | 0.72 ± 0.02 | 0.83 ± 0.02 | 1.04 ± 0.14 | 7.69 ± 1.10 | 41.2 ± 6.2 | 121.1 ± 31.1 | 10 h |
| La Barbera et al. [13] | **0.76** ± 0.02 | **0.85** ± 0.02 | 0.94 ± 0.13 | 7.06 ± 1.17 | 26.4 ± 5.4 | 119.1 ± 27.9 | 48 h |
| DeepDistanceTransform [14] | 0.72 ± 0.02 | 0.83 ± 0.02 | 1.05 ± 0.15 | 7.66 ± 1.22 | 42.0 ± 6.8 | **105.9** ± 24.6 | 18 h |
| Cascaded U-Net (prop. meth.) | 0.75 ± 0.02 | 0.84 ± 0.02 | **0.92** ± 0.13 | 6.89 ± 1.06 | **20.8** ± 3.9 | 132.6 ± 29.3 | 17 h (pre-training) + 12 h (fine-tuning) |

(a)

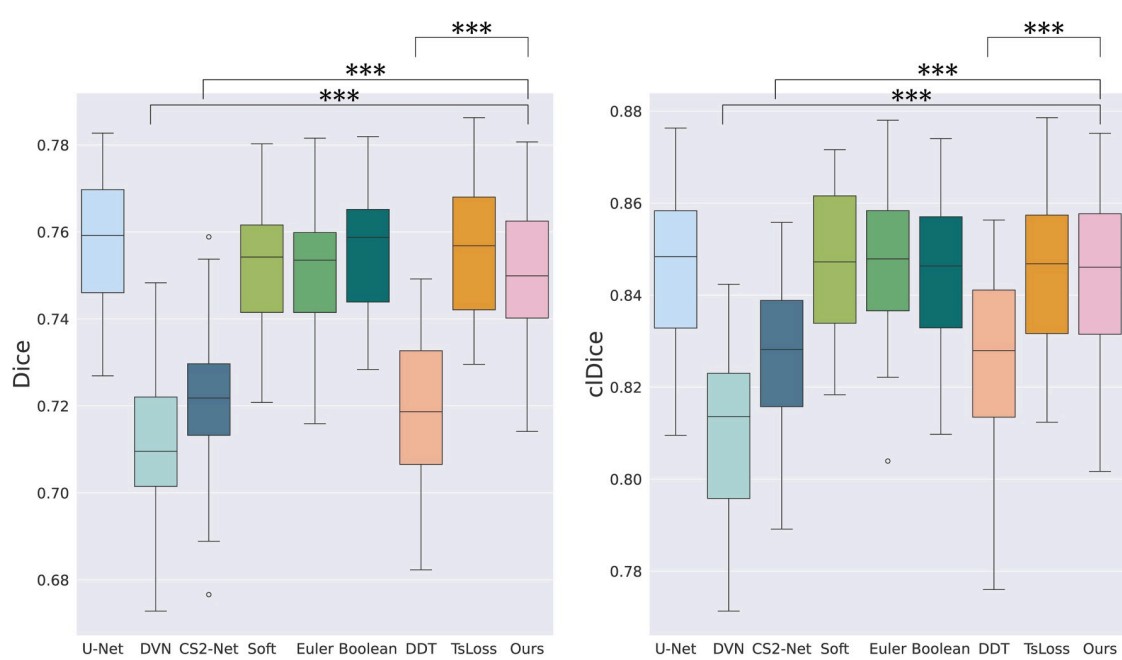

(b)

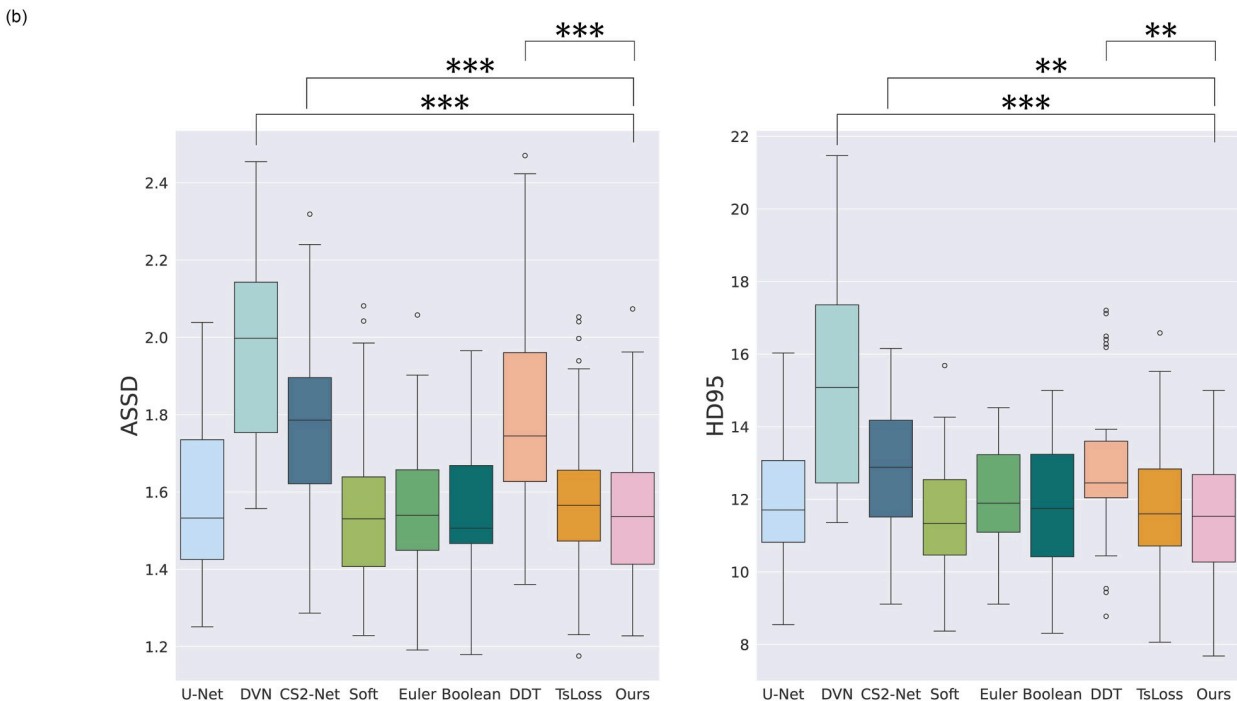

**Fig 5. Boxplots results on Bulittt dataset for (a) overlap-based metrics and (b) distance-based metrics.** Braces indicate the statistical significance between the proposed method and other methods where *** indicates p-value $\leq 0.001$, ** indicates p-value $\leq 0.01$, * indicates p-value $\leq 0.05$. No braces means that there is no statistical signifiance.

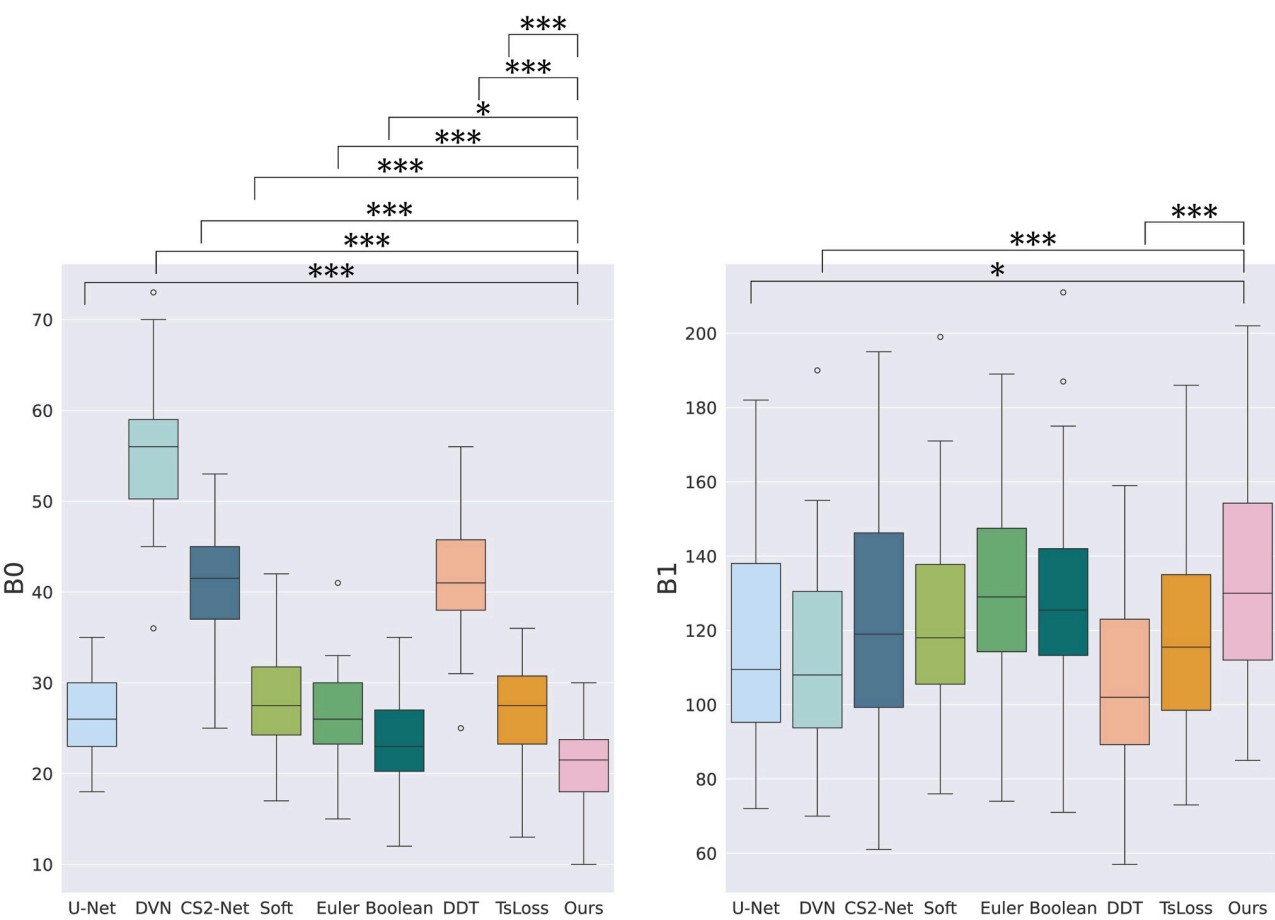

**Fig 6. Boxplots results on Bullitt dataset for topological metrics $\beta_0$ and $\beta_1$.** Braces indicate the statistical significance between the proposed method and other methods where *** indicates p-value $\leq$ 0.001, ** indicates p-value $\leq$ 0.01, * indicates p-value $\leq$ 0.05. No braces means that there is no statistical signifiance.

Furthermore, when compared to a standard U-Net and the method proposed by La Barbera et al. [13], the proposed method exhibits similar results in terms of overlap-based metrics and distance-based metrics (with differences not statistically significant). However, the proposed method performs significantly better in terms of topological metrics, as evidenced by improvements with respect to $\beta_0$ (statistically significant), highlighting its effectiveness in preserving the topology of vascular segmentation and indicating a better connectivity behavior of the proposed method.

Similar conclusions can be drawn from comparing the proposed method with those trained using the clDice with different skeletonization methods (soft-skeletonization, Euler, and Boolean). The overlap-based metrics of these methods are comparable to the proposed method (with no statistical differences); however there is a notable difference in $\beta_0$, indicating that the segmentations produced by the proposed method are significantly more connected.

Finally, note that the Boolean approach, the second most connected segmentation results, is very expensive to train due to the high runtime of the skeletonization method.

The results obtained on the IXI dataset, presented in Table 3, complement the findings on the Bullitt dataset, confirming similar trends. However, due to the limited test set of 15 patients, differences between metrics are generally not statistically significant. Our method

**Table 3. Evaluation of the compared methods on IXI dataset: Mean ± standard deviation values.**

| Model | $DSC\uparrow$ | clDice $\uparrow$ | $ASSD\downarrow$ | $HD95\downarrow$ | $\beta_0\downarrow$ | $\beta_1\downarrow$ | Training time (h) $\downarrow$ |
|---|---|---|---|---|---|---|---|
| U-Net (Dice) | **0.84** ± 0.03 | **0.88** ± 0.02 | 0.45 ± 0.14 | 3.13 ± 2.35 | 25.9 ± 7.2 | 64.1 ± 17.6 | 17 h |
| U-Net (Dice + clDice Soft) [10] | 0.78 ± 0.03 | 0.86 ± 0.04 | 0.64 ± 0.28 | 4.28 ± 2.98 | 31.3 ± 7.4 | 70.7 ± 26.4 | 18 h |
| U-Net (Dice + clDice Euler) [11] | 0.83 ± 0.03 | **0.88** ± 0.03 | 0.47 ± 0.21 | 3.39 ± 3.04 | 27.4 ± 7.2 | 69.4 ± 15.3 | 28 h |
| U-Net (Dice + clDice Boolean) [11] | 0.83 ± 0.03 | **0.88** ± 0.02 | 0.45 ± 0.18 | 3.18 ± 2.70 | 25.3 ± 6.1 | 73.0 ± 19.3 | 42 h |
| DeepVesselNet [16] | 0.81 ± 0.03 | 0.86 ± 0.03 | 0.70 ± 0.37 | 6.03 ± 5.41 | 31.3 ± 9.6 | 49.6 ± 12.1 | 10 h |
| CS2-Net [15] | 0.82 ± 0.02 | 0.87 ± 0.03 | 0.54 ± 0.27 | 3.99 ± 3.43 | 28.1 ± 6.4 | 52.9 ± 13.2 | 10 h |
| La Barbera et al. [13] | **0.84** ± 0.03 | **0.88** ± 0.03 | 0.44 ± 0.17 | 2.96 ± 2.50 | 26.6 ± 7.3 | 64.9 ± 17.4 | 48 h |
| DeepDistanceTransform [14] | 0.81 ± 0.02 | 0.87 ± 0.02 | 0.56 ± 0.22 | 4.19 ± 2.54 | 27.1 ± 7.6 | **45.1** ± 10.8 | 18 h |
| Cascaded U-Net (prop. meth.) | 0.83 ± 0.03 | **0.88** ± 0.026 | **0.42** ± 0.16 | **2.79** ± 2.51 | **24.3** ± 5.8 | 69.5 ± 20.8 | 17 h (pre-training) + 12 h (fine-tuning) |

outperforms CS2-Net [15], DeepVesselNet [16] and DeepDistanceTransform [14] in $\beta_0$ and distance-based metrics, with slight differences in overlap-based metrics. Additionally, the proposed method outperforms the standard U-Net trained with a Dice loss and the method proposed by La Barbera et al. [13] in terms of $\beta_0$. Finally, when comparing to the clDice-based methods with different skeletonization algorithms, the proposed method still exhibits the lowest $\beta_0$ value, indicating superior connectivity behavior, and is more cost-effective to train.

Beyond this quantitative analysis, it is important to investigate the results from a qualitative point of view. In particular, the good reconnection behavior induced by clDice and cascaded U-Net is observed on both datasets in Figs 7 and 8. Firstly, it can be observed that many disconnections are presented by the U-Net trained with either the standard Dice loss or the clDice loss with the soft-skeleton algorithm. The best connectivity is observed visually for Boolean and the proposed method, confirming the interest of a topologically accurate skeleton when using the clDice loss. The poor connectivity behavior of DeepVesselNet, CS2-Net, *TsLoss* and DeepDistanceTransform methods is also visible on both datasets.

## 5 Discussion

In this section, some results and findings of the study are first discussed. Following that, prevalent limitations inherent in the work are addressed.

Regarding $\beta_1$, as discussed in Section 4.2 none of the compared method constrains the number of tunnels. Also all methods except DeepDistanceTransform present a high value with respect to the expected theoretical value ($\beta_1 = 1$), showing that they all fail to really preserve the topology in that aspect.

An interesting finding of this study is that despite that the skeletons produced by the proposed method exhibit lower topological accuracy than the Euler and Boolean methods, the segmentation results present a superior connectivity. This good behavior is attributed to two key factors. First, unlike deterministic methods, the proposed learned skeletonization model can correct segmentation errors during training (notably leveraging information from the MRA image), thus mitigating error propagation. Second, the proposed skeletonization acts as a complementary task to the segmentation. As a result of this joint multitask learning approach, the segmentation task can benefit from an inductive bias to learn a topologically accurate segmentation.

It is also convenient to comment further on the results obtained by La Barbera et al. and DeepDistanceTransform methods. DeepDistanceTransform relies on the geometry refinement of the segmentation from the distance map generated by the model. This distance map is essentially a multiclass classification task, where the ground truth is a quantized version of the

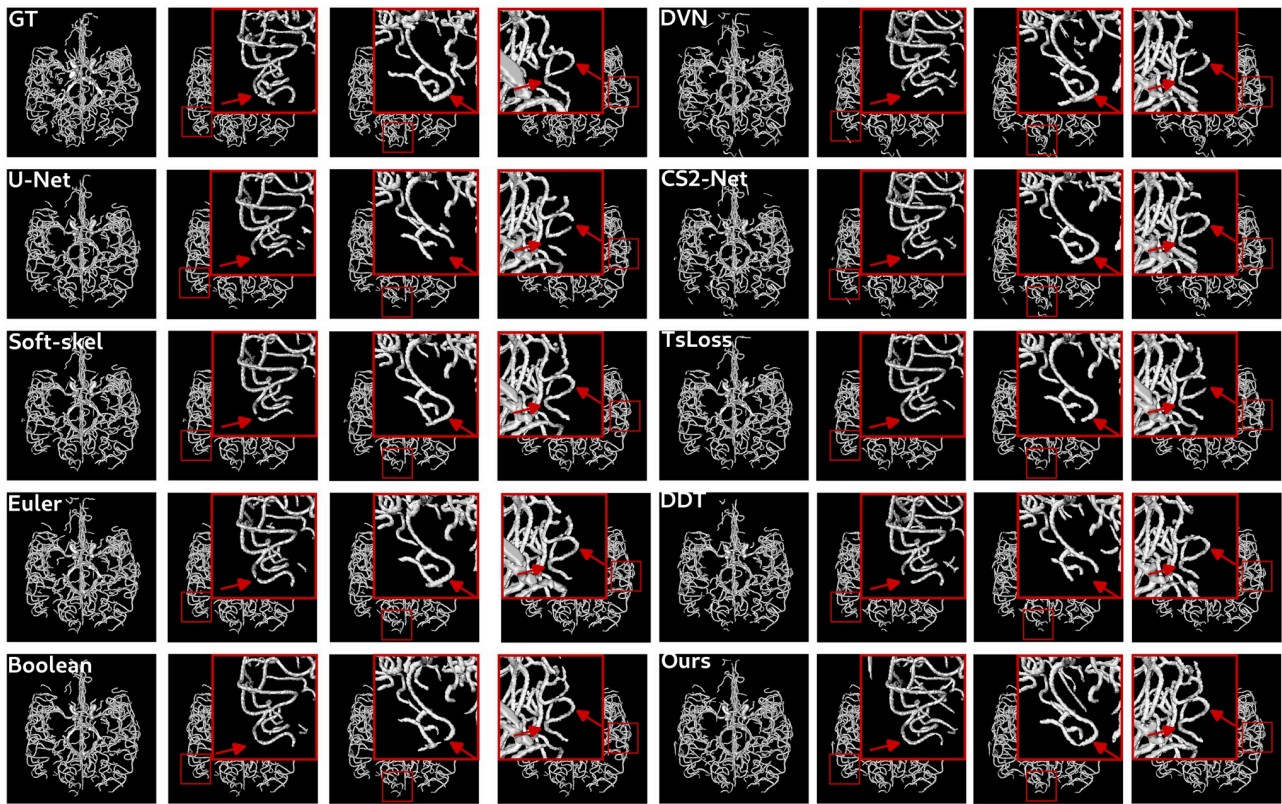

**Fig 7. Segmentation results obtained with the different methods for patient 8 of Bullitt dataset.** Three zoomed area highlighted with red boxes are presented to observe better the connectivity behavior of each method. Red arrows indicate interesting areas which can present misconnections.

true distance map. However, this task is inherently unbalanced as background voxels are predominant and large distance map values are rare (*i.e.* large vessels). Consequently, accurately learning this distance map becomes challenging. In the experiments, it was observed that the distance values were generally overestimated, leading to errors in the final segmentation. Nevertheless, this method appears to be the only one capable of efficiently reducing the number of tunnels, thanks to the geometry-aware refinement method.

La Barbera et al. method relies on a topological loss enforced by the Frangi vesselness and morphological similarity losses. The first one encourages voxels belonging to the manual annotation to have a high vesselness value, while the second compares the eigen values of the Hessian matrix of the predicted segmentation and the manual annotation. In the conducted experiments, this approach yields similar results to the 3D U-Net trained with a standard Dice loss, contrary to what is observed in [13]. This divergence may be explained by the fact that in the experiments, the brain vascular network exhibits a more complex geometry with a higher number of bifurcations, compared to the initial study carried out in [13]; the Frangi vesselness is well-known to yield suboptimal results around bifurcations [17]. Also this method requires multiple parameters tuning and it is likely that a better parameter combinaison can be found for this case study through an exhaustive optimization.

Our study also highlights some limitations regarding the evaluation of cerebrovascular segmentation which are important to emphasize. First, the lack of large annotated public datasets restricts the impact of drawing generalizable conclusions with statistical significance. This is particularly noticeable with the IXI dataset. Second, vascular segmentations are very difficult

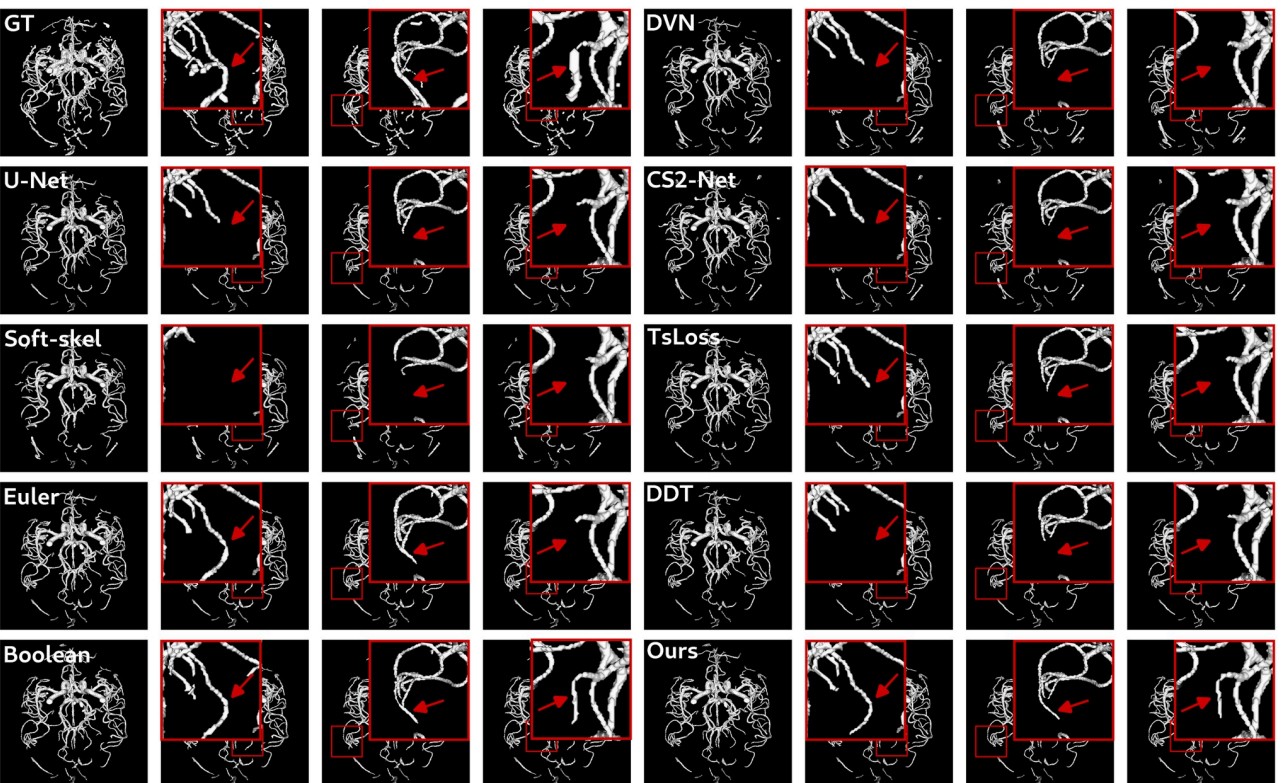

**Fig 8. Segmentation results obtained with the different methods for patient IXI023 of IXI dataset.** Three zoomed area highlighted with red boxes are presented to observe better the connectivity behavior of each method. Red arrows indicate interesting areas which can present misconnections.

to evaluate quantitatively. Classic overlap-based metrics are indeed not well suited for this evaluation, since an offset error of only a few voxels may cause a mismatch between the segmentation and the manual annotation and then lead to a drop in the metrics. By contrast, distance based metrics or specific metrics like clDice [10] are less sensitive to such kinds of errors, and can then complement the usual overlap based metrics. Nonetheless, these metrics also fail in assessing the correct topology of the segmentations.

The Betti numbers characterize the topology of an object. However this evaluation is independent of the manual annotation. Two segmentations may have the same Betti numbers (*i.e.* a similar topology), but exhibit very different geometries. Moreover, the annotations were built without particular considerations regarding the topology of the segmentation. This is why these references are most often topologically imperfect and present large Betti numbers [36]. Thus although it is common to compute the difference between Betti numbers of the segmentation and the manual annotation, this is not always relevant since the underlying assumption is that the topology of the manual annotation is correct.

## 6 Conclusion

In this article, it was proposed to use a U-Net to learn the skeletonization operation required to compute the clDice. This method provides a good trade-off between the required topological correctness of the skeletons and the computation time. A cascaded multitask U-Net was then proposed to learn vascular segmentation with topological guidance modeled via the

clDice loss. This cascaded multitask U-Net jointly learns vessel segmentation and skeletonization and can then benefit from the inductive bias induced by the skeletonization task.

The proposed approach was compared to state-of-the-art methods for vascular segmentation applied to cerebrovascular structures on two publicly available TOF-MRA datasets. Specifically, two topological losses for vascular segmentation were compared: *TsLoss* from La Barbera et al. [13] and clDice from Shit et al. [10] with several skeletonization algorithms. All these approaches were reimplemented in a common PyTorch framework, which is publicly available, to promote further developments of new topology-aware methods for cerebrovascular segmentation.

In this study, it has been demonstrated that the clDice loss improves the topological correctness of cerebrovascular segmentation from MRA images. Moreover, the proposed method has been shown to improve the topological correctness of vascular segmentation with a lower training time.

There are opportunities for further enhancement of the proposed method. Investigating dedicated skeletonization network architectures or improving the information sharing between the segmentation and skeletonization tasks are potential avenues for refining the proposed approach. Additionally, it would be valuable to compare the methods on datasets of patients with cardiovascular pathologies to evaluate the robustness and generalizability of the proposed approach. These aspects will be the focus of future research efforts.

## Acknowledgments

This work was granted access to the HPC resources of IDRIS under the allocation 2022-AD011013610 made by GENCI. We would like to thank Maria A. Zuluaga and Francesco Galati for graciously providing the IXI annotations.

## Declaration of generative AI and AI-assisted technologies in the writing process

During the preparation of this work, the authors used ChatGPT in order to propose reformulation of some phrases. After using this tool/service, the authors reviewed and edited the content as needed and take full responsibility for the content of the publication.

## Author Contributions

**Conceptualization:** Pierre Rougé, Odyssée Merveille.

**Formal analysis:** Pierre Rougé.

**Funding acquisition:** Nicolas Passat, Odyssée Merveille.

**Investigation:** Pierre Rougé.

**Methodology:** Pierre Rougé.

**Project administration:** Nicolas Passat.

**Supervision:** Nicolas Passat, Odyssée Merveille.

**Validation:** Pierre Rougé, Nicolas Passat, Odyssée Merveille.

**Visualization:** Pierre Rougé.

**Writing – original draft:** Pierre Rougé, Odyssée Merveille.

**Writing – review & editing:** Pierre Rougé, Nicolas Passat, Odyssée Merveille.

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
