## [Decision Letter · Decision Letter 0]

28 Jun 2024

PONE-D-24-21887Topology Aware Multitask Cascaded U-Net for Cerebrovascular SegmentationPLOS ONE

Dear Dr. Rougé,

Thank you for submitting your manuscript to PLOS ONE. After careful consideration, we feel that it has merit but does not fully meet PLOS ONE’s publication criteria as it currently stands. Therefore, we invite you to submit a revised version of the manuscript that addresses the points raised during the review process.

Please carefully read and address the comments from the reviewers provided below.

We look forward to receiving your revised manuscript.

Kind regards,

Tomo Popovic, Ph.D.

Academic Editor

PLOS ONE

Journal Requirements:

   "Funding by the Agence National de la Recherche under the grant (ANR-20-CE45-0011)."

4. For studies involving third-party data, we encourage authors to share any data specific to their analyses that they can legally distribute. PLOS recognizes, however, that authors may be using third-party data they do not have the rights to share. When third-party data cannot be publicly shared, authors must provide all information necessary for interested researchers to apply to gain access to the data. (https://journals.plos.org/plosone/s/data-availability#loc-acceptable-data-access-restrictions) 

Reviewers' comments:

Reviewer's Responses to Questions

**Comments to the Author**

1. Is the manuscript technically sound, and do the data support the conclusions?

Reviewer #1: Partly

Reviewer #2: Partly

2. Has the statistical analysis been performed appropriately and rigorously? 

Reviewer #1: I Don't Know

Reviewer #2: Yes

3. Have the authors made all data underlying the findings in their manuscript fully available?

Reviewer #1: Yes

Reviewer #2: Yes

4. Is the manuscript presented in an intelligible fashion and written in standard English?

Reviewer #1: Yes

Reviewer #2: Yes

5. Review Comments to the Author

Reviewer #1: This research paper addresses the critical topic of automatic 3D segmentation of the cerebrovascular network. It employs a U-Net model trained using Dice and centerline Dice losses, resulting in reduced training time while better preserving the network's topology compared to other known methods for cerebrovascular segmentation. While the topic is very important and the approach the authors use is innovative, the manuscript suffers from a critical deficiency that needs correction.

Major correction:

1. Questioning the quality of the ground truth, which normally used as a gold standard.

In line 294, it is stated that the value of is expected to be B1 = 1 and the value of B0 = 1. The authors continue with the explanation that the ground truths are noisy and topologically incorrect, so the actual numbers they measured and presented in the results section are much higher (they are not even the same order of the magnitude). How do the authors evaluate what is correct if the gold standard test is not correct? That puts into question all the conclusions in the paper and, therefore, this section needs to be re-written.

Related to this, what is the unit of measurement or scale on the Y-axis in Figures 7 and 8?

Minor suggestions:

The presented work needs some additional information regarding the data sets used to test this newly developed method for cerebrovascular segmentation. Including the additional information would underscore the strengths and the true value of this research in real-world clinical applications.

1. Inter/Expert Segmentation Variability.

The ground truth segmentations were performed by experts. Is there any information from previously published data on the expert-to-expert (human-to-human) variability of segmentations? Specifically, are the observed differences between expert-to-expert segmentation results comparable to the differences measured in this study (newly described automatic method vs. other state-of-the-art automatic methods, i.e. automatic- to -automatic method comparison)?

2. Data Characteristics.

Was the method tested on a dataset from healthy patients, or was cerebrovascular pathology present? Pathological conditions such as leaky vessels, inflammatory cell infiltrates, and bleeding could interfere with segmentation and evaluation, potentially leading to decreased accuracy and increased training time. Although all methods compared in this study used the same datasets, and therefore the observed differences in the results are not caused by differences in the health status of the cerebrovascular network, mentioning this fact is important. It demonstrates that the newly described method can be utilized in various situations normally encountered in real life, and not only under certain conditions (for example, only in healthy young people, because that would defeat the purpose of developing this method in the first place).

Technical comments:

1.Organization of the Manuscript.

The results section should contain only the results, referencing appropriate tables and figures after each section or statement. Comparisons to other works and conclusions should be moved to the discussion section.

2. Consistency in the Paper Organization and Presentation.

The methods section explains B0, B1, and B2 numbers, while the results figures contain B1 and B0 numbers. However, the authors continue to discuss only B0. The reason for this needs better clarification and consistency.

3. Abbreviations.

Add a list of abbreviations (e.g., MRA, CTA, TOF-MRA) or define the meaning of each abbreviation in parentheses the first time it is mentioned in the text.

Reviewer #2: This article examines the topic of automatic 3D cerebrovascular network segmentation using a cascaded U-net model, incorporating a novel method for evaluating segmentation loss with Dice losses. Researchers trained and evaluated their U-Net model in comparison with other leading skeletonization models, with a controlled experimental setting. The authors report improved accuracy and faster prediction times. In addition to its primary contribution, this research also offers additional value by proposing a more effective computation of Dice loss and providing an open code repository for further research access.

The explanation of the proposed U-net model is sound and very well documented.

The research's purpose, methods, and experiments are well articulated, but the descriptions of the dataset(subsection 4.1) and metrics (subsection 4.2) have drawbacks that raise concerns on validity of achieved results.

1. Based on statements on Betti number values for ground truth data “This shows that the ground-truth are noisy and not topologically correct” - This raises a significant challenge in evaluating segmentation methods; doesn’t this directly affect the withdrawn conclusions of the study and raises questions on model’s performance?

2. Regarding the omission of β2 results due to no apparent cavities, could you clarify how this was determined? Was the absence of cavities verified visually, mathematically, or through a combination of both methods?

3.The dataset description in subsection 4.1 is rather short. The study cited for the publicly available dataset does not contain a reference for the dataset itself; it would be beneficial to include a direct link to the repository containing the dataset along with the citation. It is unclear whether the vessel segmentation was performed by one or more experts. If done by multiple experts, what about consistency of the segmentations (e.g., expert to expert comparison)? Subsection 4.1 notes that the skeletonize method from scikit-image was used for generating the skeleton ground truth, but does not discuss why this method was chosen over other existing methods (such as graph based etc.).

Additionally, the written language of the article can be enhanced by using an objective and formal tone, replacing first person pronouns (we, ours) with passive voice constructions.

6. PLOS authors have the option to publish the peer review history of their article (what does this mean?). If published, this will include your full peer review and any attached files.

Reviewer #1: No

Reviewer #2: No

---

## [Author Response · Author response to Decision Letter 0]

18 Aug 2024

Rebuttal Letter

We would like to warmly thank the Editor for the management of the review

process of this manuscript. We also thank the Reviewers for having taken the

time to carefully study our manuscript and for their valuable comments and

suggestions that induced improvements of the manuscript. A point-by-point

response to each of the issues raised by the Reviewers is given in this document.

Reviewer #1

Reviewer:

This research paper addresses the critical topic of automatic 3D segmentation of

the cerebrovascular network. It employs a U-Net model trained using Dice and

centerline Dice losses, resulting in reduced training time while better preserving

the network’s topology compared to other known methods for cerebrovascular

segmentation. While the topic is very important and the approach the authors

use is innovative, the manuscript suffers from a critical deficiency that needs

correction.

Major correction

#1 Questioning the quality of the ground truth, which normally used as a

gold standard. In line 294, it is stated that the value of is expected to be B1

= 1 and the value of B0 = 1. The authors continue with the explanation that

the ground truths are noisy and topologically incorrect, so the actual numbers

they measured and presented in the results section are much higher (they are not

even the same order of the magnitude).

How do the authors evaluate what is correct if the gold standard test is not

correct? That puts into question all the conclusions in the paper and, therefore,

this section needs to be re-written.

Authors:

We thank the two Reviewers for pointing out this issue and providing us with

the opportunity to clarify. We acknowledge that the “ground truths” we em-

ployed are indeed noisy and topologically incorrect. To address this, we have

replaced the term “ground truth” with “manual annotation” in the manuscript

and provided a more detailed explanation as follows: “Unlike other metrics,

Betti numbers characterize the topology of a structure independently of its pro-

posed annotation. Analyzing the Betti numbers of a segmentation results then

requires comparing them with the true (ground truth) Betti numbers. From

an anatomical point of view, the topology of the brain arterial network is well

established. All arteries are connected, thus β0 is equal to 1; there is one tunnel

(the circle of Willis), thus β1 is equal to 1; and no cavities are present, thus β2

is equal to 0. However, the Betti numbers of the Bullitt and IXI annotations

were computed, and much greater values for β0 and β1 were observed (see Table

1). This discrepancy is a typical issue with segmentation annotations of geo-

metrically complex 3D structures. Manual annotation is a labor-intensive task,

usually performed on 2D slices, making it difficult to accurately capture the 3D

geometry and topology of the annotated structure. Moreover, the annotation

guidelines rarely focus on topology, as voxel-wise metrics are the gold standard

for segmentation evaluation. This leads to annotations that are suitable for

voxel-wise analysis but often inadequate for topological analysis. Our goal is

to obtain vascular segmentation that reflects the true topology of the brain ar-

terial network. In the following, the ground-truth Betti numbers of the brain

artery network (β0 = 1, β1 = 1, and β2 = 0) are used as the reference for the

segmentation Betti numbers.”

Reviewer:

Related to this, what is the unit of measurement or scale on the Y-axis in Figures

7 and 8 ?

Authors:

The Dice and ClDice are dimension-less and then without unit. They are also

normalized metrics (between 0 and 1). The Betti numbers are the ranks of the

successive homology groups. They are also given without unit, whereas they

represent the numbers of connected components, tunnels and cavities, respec-

tively. In the initial version of the manuscript, HD95 and ASSD (which are

distance measures) were initially expressed in voxels. We consider that it would

be more appropriate to express them in millimeters due to the non-isotropic

nature of the images (voxel dimensions: 0.5 × 0.5 × 0.8 mm3). As a conse-

quence, we have recomputed the HD95 and ASSD in millimeters and modified

the manuscript accordingly. This adjustment does not induce a significative

change in the results nor alter the conclusions stated in the manuscript.

Reviewer:

Minor suggestions

The presented work needs some additional information regarding the data

sets used to test this newly developed method for cerebrovascular segmentation.

Including the additional information would underscore the strengths and the true

value of this research in real-world clinical applications.

#1 Inter/Expert Segmentation Variability.

The ground truth segmentations were performed by experts. Is there any

information from previously published data on the expert-to-expert (human-to-

human) variability of segmentations? Specifically, are the observed differences

between expert-to-expert segmentation results comparable to the differences mea-

sured in this study (newly described automatic method vs. other state-of-the-art

automatic methods, i.e. automatic-to-automatic method comparison)?

Authors:

We thank the reviewer for highlighting a major difficulty in evaluating cere-

brovascular segmentation methods. The annotation of blood vessels in 3D im-

ages is a very tedious and error-prone task. Even with semi-automatic tools, an

expert takes several hours to annotate an image. This explains why there are

very few publicly available datasets, particularly for cerebral vascular segmen-

tation, which involves a very complex network. To the best of our knowledge,

no publicly available cerebrovascular dataset provides annotations from more

than one expert per volume. Consequently, it is difficult to evaluate the expert-

to-expert variability quantitatively. We agree that this question is central and

we are currently working on addressing it. However, we believe that it is out

of the scope of the current article. The reviewer may look at the following

preprint where we discuss the origin of this expert-to-expert variability and

propose guidelines to reduce it: https://arxiv.org/abs/2404.01765.

Reviewer:

#2 Data Characteristics.

Was the method tested on a dataset from healthy patients, or was cerebrovas-

cular pathology present? Pathological conditions such as leaky vessels, inflam-

matory cell infiltrates, and bleeding could interfere with segmentation and eval-

uation, potentially leading to decreased accuracy and increased training time.

Although all methods compared in this study used the same datasets, and there-

fore the observed differences in the results are not caused by differences in the

health status of the cerebrovascular network, mentioning this fact is important.

It demonstrates that the newly described method can be utilized in various situ-

ations normally encountered in real life, and not only under certain conditions

(for example, only in healthy young people, because that would defeat the purpose

of developing this method in the first place).

Authors:

We thank the reviewer for raising a valid concern regarding our study. Both

datasets we used consist solely of healthy subjects. We agree that it would

be very interesting to compare the methods on patients with vascular diseases.

However, to the best of our knowledge, there are no publicly available datasets

with segmentation annotations for pathological patients that would allow us to

conduct such an analysis.

We are currently working on an annotation protocol for the segmentation of

brain arteries on a dataset of patients with ischemic stroke. In the meantime,

we believe that working on healthy subjects is relevant because most of the

brain arteries in pathological patients (e.g., patients with stenosis or aneurysms)

are similar to those in healthy patients. The question of vascular connectivity

remains crucial for patients with cardiovascular pathologies, although additional

challenges will arise. We have added the following sentence to the future work

discussion of the conclusion of the article: “Additionally, it would be valuable

to compare the methods on datasets of patients with cardiovascular pathologies

to evaluate the robustness and generalizability of the proposed approach.”

Reviewer:

Technical comments.

#1 Organization of the Manuscript. The results section should contain

only the results, referencing appropriate tables and figures after each section or

statement. Comparisons to other works and conclusions should be moved to the

discussion section.

Authors:

We thank the reviewer for this comment. We have reorganized the manuscript

to make the separation between the results and discussion sections clearer.

Reviewer:

#2 Consistency in the Paper Organization and Presentation.

The methods section explains B0, B1, and B2 numbers, while the results

figures contain B1 and B0 numbers. However, the authors continue to discuss

only B0. The reason for this needs better clarification and consistency.

Authors:

We thank the two reviewers for their insightful comments on that point. In

3D, the third Betti number β2 represents the number of cavities in the studied

structure. Anatomically, a vascular network (and especially the flowing blood

that is visualized in angiographic images) does not contain cavities in healthy

subjects and in absence of acquisition artifacts, so β2 should be equal to 0.

In practice, we computed the β2 scores for all segmentation results and we

consistently obtained β2 = 0 across all cases.

Given that β2 did not provide additional useful information and was uni-

formly zero, we decided to remove this column from the result tables. We have

also clarified this point in the metrics section of the manuscript: “Additionally,

the value of β2 was computed for both Bullitt and IXI datasets, it was observed

that the values were null for both manuals annotations and predicted segmen-

tations (corresponding to the theoretical value). Therefore, the β2 values are

not presented or discussed in the following sections.”

In the following, we complete our answer to explain the lack of analysis of

β1 results.

In the literature, research focused on improving the topology of vascular seg-

mentations typically emphasizes enhancing connectivity (β0). This emphasis is

due to β0 being the most crucial and interpretable parameter for subsequent

clinical and research applications. Conversely, tunnels (β1) are much harder

to visualize and interpret and methods usually do not attempt to reduce them

directly in segmentations. Additionally, they are more prevalent in the annota-

tions, which we believe is primarily due to annotation noise rather than genuine

topological errors. For these reasons, it did not appear relevant to analyze the

β1 results. However, we have decided to include them in our study to maintain

transparency and provide a complete picture of our findings.

Reviewer:

#3 Abbreviations. Add a list of abbreviations (e.g., MRA, CTA, TOF-MRA)

or define the meaning of each abbreviation in parentheses the first time it is

mentioned in the text.

Authors:

We have carefully reviewed the manuscript to ensure that each abbreviation is

clearly defined upon its first mention.

Reviewer #2

Reviewer:

This article examines the topic of automatic 3D cerebrovascular network seg-

mentation using a cascaded U-net model, incorporating a novel method for eval-

uating segmentation loss with Dice losses. Researchers trained and evaluated

their U-Net model in comparison with other leading skeletonization models, with

a controlled experimental setting. The authors report improved accuracy and

faster prediction times. In addition to its primary contribution, this research

also offers additional value by proposing a more effective computation of Dice

loss and providing an open code repository for further research access.

The explanation of the proposed U-net model is sound and very well docu-

mented.

The research’s purpose, methods, and experiments are well articulated, but

the descriptions of the dataset (subsection 4.1) and metrics (subsection 4.2) have

drawbacks that raise concerns on validity of achieved results.

#1 Based on statements on Betti number values for ground truth data “This

shows that the ground-truth are noisy and not topologically correct” - This raises

a significant challenge in evaluating segmentation methods; doesn’t this directly

affect the withdrawn conclusions of the study and raises questions on model’s

performance?

Authors:

See answer to Reviewer 1 regarding the same question.

Reviewer:

#2 Regarding the omission of β2 results due to no apparent cavities, could you

clarify how this was determined? Was the absence of cavities verified visually,

mathematically, or through a combination of both methods?

Authors:

We thank the reviewer for allowing us to clarify this point. We have checked the

absence of cavities in the segmentation results computationally. We have com-

puted the β2 scores for each segmentation result from each compared approach

and obtained consistently a value equal to 0. We believe that it is not possible to

evaluate the absence of cavities on a segmentation results because of the number

and spreading of the vessels in the image and the difficulty of visualizing a 3D

structure on a 2D screen. We have clarified this point in the metric section:

“Additionally, the value of β2 was computed for both Bullitt and IXI datasets,

it was observed that the values were null for both manuals annotations and pre-

dicted segmentations (corresponding to the theoretical value). Therefore, the

β2 values are not presented or discussed in the following sections”

Reviewer:

#3 The dataset description in subsection 4.1 is rather short. The study cited for

the publicly available dataset does not contain a reference for the dataset itself;

it would be beneficial to include a direct link to the repository containing the

dataset along with the citation. It is unclear whether the vessel segmentation

was performed by one or more experts. If done by multiple experts, what about

consistency of the segmentations (e.g., expert to expert comparison)?

Authors:

We thank the reviewer for this insightful comment. For this study, we used two

publicly available datasets: Bullitt1 [3] and IXI2. We have provided the links to

the web page describing each dataset in the Datasets section of the manuscript.

We have also clarified that the annotation comes from a unique expert for each

volume. Unfortunately, as explained in our answers to Reviewer #1, there is

currently no publicly available dataset with annotations from several experts.

1https://public.kitware.com/Wiki/TubeTK/Data

2https://brain-development.org/ixi-dataset/

This limitation prevents us from analyzing our results with respect to inter-

expert variability.

Reviewer:

Subsection 4.1 notes that the skeletonize method from scikit-image was used for

generating the skeleton ground truth, but does not discuss why this method was

chosen over other existing methods (such as graph based etc.).

Authors:

We chose the Lee skeletonization algorithm [2] because it is the gold standard

used in the vascular segmentation community [4, 5]. It is known to be an

algorithm that provides accurate skeletons results. We have clarified this point

in the manuscript in the Datasets section.

Reviewer:

Additionally, the written language of the article can be enhanced by using an ob-

jective and formal tone, replacing first-person pronouns (we, ours) with passive

voice constructions.

Authors:

We appreciate the reviewer’s suggestion for improving the language of the

manuscript. In response, we have revised the manuscript to adopt a more

objective and formal tone by replacing first-person pronouns with passive voice

constructions where appropriate.

References

[1] Rougé P, Conze PH, Passat N, Merveille O. Guidelines for Cerebrovascular

Segmentation: Managing Imperfect Annotations in the context of Semi-

Supervised Learning. arXiv preprint arXiv:240401765. 2024;.

[2] Le

---

## [Decision Letter · Decision Letter 1]

19 Sep 2024

Topology Aware Multitask Cascaded U-Net for Cerebrovascular Segmentation

PONE-D-24-21887R1

Dear Dr. Rougé,

We’re pleased to inform you that your manuscript has been judged scientifically suitable for publication and will be formally accepted for publication once it meets all outstanding technical requirements.

Kind regards,

Tomo Popovic, Ph.D.

Academic Editor

PLOS ONE

Reviewers' comments:

Reviewer's Responses to Questions

**Comments to the Author**

1. If the authors have adequately addressed your comments raised in a previous round of review and you feel that this manuscript is now acceptable for publication, you may indicate that here to bypass the “Comments to the Author” section, enter your conflict of interest statement in the “Confidential to Editor” section, and submit your "Accept" recommendation.

Reviewer #1: All comments have been addressed

Reviewer #2: All comments have been addressed

2. Is the manuscript technically sound, and do the data support the conclusions?

Reviewer #1: Yes

Reviewer #2: Yes

3. Has the statistical analysis been performed appropriately and rigorously? 

Reviewer #1: Yes

Reviewer #2: Yes

4. Have the authors made all data underlying the findings in their manuscript fully available?

Reviewer #1: Yes

Reviewer #2: (No Response)

5. Is the manuscript presented in an intelligible fashion and written in standard English?

Reviewer #1: Yes

Reviewer #2: Yes

6. Review Comments to the Author

Reviewer #1: (No Response)

Reviewer #2: Thank you for adressing the comments and revising the article. The changes have significantly improved the clarity and quality of the manuscript. The responses to reviewer comments were very insightful.

7. PLOS authors have the option to publish the peer review history of their article (what does this mean?). If published, this will include your full peer review and any attached files.

Reviewer #1: No

Reviewer #2: **Yes: **Dejan Babic

---

## [Editor Report · Acceptance letter]

27 Sep 2024

PONE-D-24-21887R1 

PLOS ONE

Dear Dr. Rougé, 

I'm pleased to inform you that your manuscript has been deemed suitable for publication in PLOS ONE. Congratulations! Your manuscript is now being handed over to our production team.

Kind regards, 

on behalf of

Prof. Tomo Popovic 

Academic Editor

PLOS ONE